# Multi-Omics of *Campylobacter jejuni* Growth in Chicken Exudate Reveals Molecular Remodelling Associated with Altered Virulence and Survival Phenotypes

**DOI:** 10.3390/microorganisms12050860

**Published:** 2024-04-25

**Authors:** Lok Man, Pamela X. Y. Soh, Tess E. McEnearney, Joel A. Cain, Ashleigh L. Dale, Stuart J. Cordwell

**Affiliations:** 1School of Life and Environmental Sciences, The University of Sydney, Sydney, NSW 2006, Australia; 2Charles Perkins Centre, The University of Sydney, Sydney, NSW 2006, Australia; 3Sydney Mass Spectrometry, The University of Sydney, Sydney, NSW 2006, Australia

**Keywords:** biofilm, lactate, lipid A, metabolomics, motility, nutrient transport, proteomics, serine, succinate, virulence

## Abstract

*Campylobacter jejuni* is the leading cause of foodborne human gastroenteritis in the developed world. Infections are largely acquired from poultry produced for human consumption and poor food handling is thus a major risk factor. Chicken exudate (CE) is a liquid produced from defrosted commercial chicken products that facilitates *C. jejuni* growth. We examined the response of *C. jejuni* to growth in CE using a multi-omics approach. Changes in the *C. jejuni* proteome were assessed by label-based liquid chromatography coupled with tandem mass spectrometry (LC-MS/MS). We quantified 1328 and 1304 proteins, respectively, in experiments comparing 5% CE in Mueller–Hinton (MH) medium and 100% CE with MH-only controls. These proteins represent 81.8% and 80.3% of the predicted *C. jejuni* NCTC11168 proteome. Growth in CE induced profound remodelling of the proteome. These changes were typically conserved between 5% and 100% CE, with a greater magnitude of change observed in 100% CE. We confirmed that CE induced *C. jejuni* biofilm formation, as well as increasing motility and resistance against oxidative stress, consistent with changes to proteins representing those functions. Assessment of the *C. jejuni* metabolome showed CE also led to increased intracellular abundances of serine, proline, and lactate that were correlated with the elevated abundances of their respective transporters. Analysis of carbon source uptake showed prolonged culture supernatant retention of proline and succinate in CE-supplemented medium. Metabolomics data provided preliminary evidence for the uptake of chicken-meat-associated dipeptides. *C. jejuni* exposed to CE showed increased resistance to several antibiotics, including polymyxin B, consistent with changes to tripartite efflux system proteins and those involved in the synthesis of lipid A. The *C. jejuni* CE proteome was also characterised by very large increases in proteins associated with iron acquisition, while a decrease in proteins containing iron–sulphur clusters was also observed. Our data suggest CE is both oxygen- and iron-limiting and provide evidence of factors required for phenotypic remodelling to enable *C. jejuni* survival on poultry products.

## 1. Introduction

*Campylobacter jejuni* (*C. jejuni*) is a major zoonotic pathogen that is commonly found in avian species as a largely commensal gastrointestinal tract inhabitant. *C. jejuni* is thus considered a significant risk since contamination of carcass meat means poorly prepared and/or undercooked poultry remains the most common route for human infection in the developed world [1,2,3]. Unlike other food-associated pathogens, *C. jejuni* does not grow on food products, as the temperature range (~30–45 °C) and microaerophilic nature of the organism (requiring <10% O_2_ and ~10% CO_2_) are not commensurate with food spoilage. Despite this, human disease requires only very low numbers of infecting cells (~100–500 *C. jejuni* [4]) and survival on food products in a viable state occurs, even at low temperatures and high oxygen levels [5]. Therefore, limiting *C. jejuni* on consumer poultry is crucial; however, it is estimated that 30–95% of supermarket poultry is contaminated [6,7].

*C. jejuni* encodes functions that are considered essential for chicken colonisation and/or human infection. The most important ‘virulence’ determinant is flagellar-based motility, with helical cell shape and chemotaxis also appearing crucial [8,9,10,11,12]. The organism contains only a few adhesins, the most significant of which is CadF (*Campylobacter* adherence factor), although others such as fibronectin-like protein FlpA and lipoprotein JlpA also contribute to host colonisation [13,14]. *C. jejuni* does not contain a classical type 3 secretion system (T3SS), with the flagellar export apparatus utilised for the secretion of toxins, such as the *Campylobacter* invasion antigens (Cia) that facilitate invasion of the human gut epithelium [15,16,17]. Despite a limited genome of ~1620 genes (in the NCTC11168 strain [18]), *C. jejuni* has an unusual capacity for post-translationally modifying proteins. Such modifications include *O*-glycosylation of the flagellin subunits of the flagellar apparatus, a general *N*-linked glycosylation system that modifies ~100 membrane-associated proteins and is required for host cell binding, nitrate reductase activity, and antimicrobial resistance, along with widespread lysine acetylation on >1000 proteins that plays a role in metabolism and appears to regulate CadF binding to host fibronectin [19,20,21,22,23,24,25].

Understanding how *C. jejuni* adapts to food surfaces is crucial in finding interventions that limit human infection. Food-based model growth systems can thus play a role in enhancing our knowledge of the molecular basis for *C. jejuni* survival during food processing [5]. Such growth models include the use of autoclaved chicken meat, chicken skin pre-irradiated with UV light, and chicken exudate (CE) or ‘juice’ [26,27,28]. CE is generated as the thaw water from frozen or chilled poultry products, which is then filter-sterilised and used in isolation or as a supplement to other microbiological media [28]. Although no in-depth compositional studies are available, CE likely comprises blood/plasma proteins, salts, amino acids, carbohydrates, co-factors, and lipids capable of supporting *C. jejuni* growth. A broad analysis of CE revealed higher levels of protein and lower carbohydrate levels compared to Mueller–Hinton (MH) media [29]. This may be beneficial for *C. jejuni*, which lacks glycolysis and is considered asaccharolytic (some strains contain the *fuc* operon that allows growth on fucose [30,31]), utilising amino acids (serine, aspartate, glutamate, and proline) and organic acids as primary carbon sources instead [32,33]. Catabolism of amino, organic, and short-chain fatty acids rely on specific transporters, including SdaC (serine [34]), PutP (proline [35]), the Peb antigens and dicarboxylate transporters (aspartate/glutamate [36] and fumarate [37]), and LctP (lactate [38,39]).

As a microaerophile, *C. jejuni* contains a unique respiratory chain that allows the use of several alternative electron donors and acceptors [40]. Under oxygen limitation, the organism can utilise nitrate, nitrite, fumarate, and *N*- and *S*-oxides as electron acceptors [41]. Fumarate is reduced to succinate by MfrABE (Cj0437–Cj0439) methylmenaquinol:fumarate reductase (previously annotated as succinate dehydrogenase SdhABC [42]) and by the bidirectional fumarate reductase FrdABC [43], with the produced succinate secreted by the DcuB dicarboxylate transporter (DCT) and reused as a carbon source via uptake by the DctA DCT [37]. *C. jejuni* utilises electron donors including reduced flavin adenine dinucleotide (FADH), formate (via a membrane-associated formate dehydrogenase), hydrogen (via the NiFe hydrogenase), and α-ketoglutarate (α-KG; via the 2-oxoglutarate:acceptor oxidoreductase OorABCD), and these are important in host colonization [44]. A requirement for low oxygen levels results in reactive oxygen species (ROS) generation and *C. jejuni* is equipped with antioxidant proteins including catalase (KatA), superoxide dismutase (SodB), and alkylhydroperoxide reductase AhpC [45]. Tied to the presence of ROS is iron acquisition, which is tightly controlled despite a requirement for iron as a co-factor in many enzymes (e.g., Fe–S clusters). *C. jejuni* encodes a variety of uptake systems for iron, heme, and other micronutrients including molybdate, zinc, and tungsten [33].

Initial phenotypic characterisation of *C. jejuni* grown with CE showed that when incubated at either 5 °C or 48 °C, CE significantly increased the time that the organism remained viable [28]. The protective nature of CE may be mediated through biofilm production as the quorum-sensing gene *luxS* is elevated during CE-supplemented growth [46]. Growth in 5% CE resulted in an increase in biofilm production under both microaerobic and aerobic conditions, and electron microscopy revealed that *C. jejuni* preferentially attached to CE particulates bound to an abiotic surface rather than to the surface itself [47]. A second study confirmed that CE induces biofilm, and treatment with proteinase K inhibited biofilm formation, suggesting that structural integrity relies on peptides and proteins [29]. However, despite these data, no in-depth molecular analysis of *C. jejuni*’s response to growth in CE has yet been conducted.

In this study, *C. jejuni* NCTC11168 was grown in MH medium with and without supplementation with 5% CE, and in 100% CE alone, and subjected to a multi-omics analysis based on proteomics, intracellular and culture supernatant (CSN) metabolomics, and phenotypic analyses [48]. Our multi-omics approach defines CE-associated changes to *C. jejuni* physiology at the protein and metabolite levels. Quantifying these biomolecules during *C. jejuni* growth in CE may help determine how the organism survives on poultry food products for human consumption.

## 2. Materials and Methods

### 2.1. Bacterial Strains and Growth Conditions 

*C. jejuni* NCTC11168 was grown as previously described [49]. CE was obtained from a local commercial poultry butcher and was prepared by 20 min centrifugation at 8000× *g* at 4 °C, followed by overnight filtration through 0.22 µm pore size polyethersulfone (PES) membrane filters (Corning, New York, NY, USA). Cells from MH broth starter cultures were sub-cultured into fresh MH medium (in the presence or absence of 5% CE) or 100% CE at an initial OD_600_ of 0.1 and grown until the late exponential phase as determined by OD_600_ of 0.7. Cells were collected by centrifugation before lyophilization. 

### 2.2. Experimental Design

For discovery-based quantitative proteomics by tandem mass tag (TMT) labelling and LC-MS/MS of *C. jejuni* growth in MH medium with or without supplementation with 5% CE, *n* = 4 biological replicates were processed across two separate experiments. For comparison of MH controls with growth in 100% CE, *n* = 3 biological replicates were processed across 3 separate experiments. All processed proteomics data can be found in Data S1. For metabolomics analysis by targeted LC-MS/MS, a minimum of *n* = 6 biological replicates were processed. For all other experiments, biological and technical replicates are provided in the text.

### 2.3. Quantitative Proteomics by LC-MS/MS 

Peptide samples were prepared as described previously [49]. Briefly, lyophilized *C. jejuni* were reconstituted in 100 mM HEPES, pH 7.5, and lysed by 6 rounds of 30 s beadbeating. Proteins were precipitated in 1.8:2:2 water:methanol:chloroform, collected by centrifugation, washed twice with methanol, and solubilized in 8M guanidine-HCl, 100 mM HEPES, pH 7.6. Reduction and alkylation were performed with 10 mM dithiothreitol (DTT) and 20 mM iodoacetamide (IAA) for 1 hr each, respectively. Proteins were digested with sequencing-grade modified trypsin (Promega, Madison, WI, USA) at a protein:protease ratio of 30:1 overnight at 37 °C. Peptides were desalted by solid phase extraction using hydrophilic-lipophilic balance (HLB) cartridges (Waters, Bedford, MA, USA). Samples were labelled with 6-plex TMT (Thermo Scientific, Waltham, MA, USA). An excess label was quenched with hydroxylamine hydrochloride, following which labelled samples were pooled and desalted using HLB cartridges. TMT-labelled samples were fractionated offline by hydrophilic interaction liquid chromatography (HILIC) prior to LC-MS/MS as described [49]. Peptides were eluted from HILIC in 1 min fractions and adjoining fractions were pooled to a final concatenated 6–12 fractions and lyophilized. Peptide fractions were separated by reversed-phase chromatography over a 90 min gradient [49] in a Q Exactive Plus or Q Exactive™ HF Hybrid Quadrupole Mass Spectrometer (Thermo Scientific). The instruments were configured to perform one full scan MS (scan range 300–1650 *m/z*, resolution of 60,000, automatic gain control [AGC] of 3 × 10^6^, and a maximum ion injection time [IT] of 50 ms) with the top 10–15 precursors in fulfilment of the selection criteria (charge state 2–4, minimum intensity >9.2 × 10^4^, and dynamic exclusion window of 40 s) selected for MS/MS (scan range 200–2000 *m/z* with a fixed first mass of 110 *m/z*, resolution of 15,000, AGC of 1 × 10^6^, maximum IT of 50 ms, isolation window 1.4 *m/z*, and normalised collision energy [NCE] of 29).

### 2.4. Processing of Proteomics Mass Spectrometry Files 

Data from quantitative proteomics were processed in Proteome Discoverer (v. 2.2; Thermo Scientific) and searched against the UniProt *C. jejuni* NCTC11168 translated genome (UP000000799; organism ID 192222; release 24 May 2018; 1623 proteins) with the SequestHT algorithm. Search parameters were max. 2 missed cleavages and carbamidomethyl (C) as a fixed modification; and for variable modifications, oxidation (M), TMT-6plex (peptide *N*-term, K), and using precursor and fragment ion tolerances of 20 ppm. Peptide-level false discovery rate (FDR) was determined using Percolator (v. 2.08.01). Peptide spectral matches (PSMs) corresponding to a 1% FDR were exported, and reporter intensities normalised to total reporter ion signals across all channels. For relative quantitation and statistical analysis, data were imported into Perseus (v. 1.6.1.1) and analysed as per [25].

### 2.5. Targeted Metabolomics by LC-MS/MS 

Metabolites and their MS parameters (precursor, product ion *m/z* [transitions], collision energy, and declustering potential) are summarized in Appendix A. Cells (*n* = 6–9 biological replicates) were lysed in ultrapure water by 6 rounds of 30 s beadbeating. For assay of culture supernatant (CSN) metabolites (*n* = 6 biological replicates), aliquots were collected at 0, 4, 24, and 48 h growth, cells were removed by centrifugation, and supernatants were filtered through 0.22 μm PES filters. A total of 25 μL of each sample was added to 75 μL of extraction buffer (0.1% formic acid (FA) in 80:20 ethanol:water (*v*/*v*)), mixed by vortexing, and incubated at 4 °C for 2 h. Solutions were vortexed then centrifuged at 14,000× *g* at 4 °C for 15 min. A total of 50 μL of the supernatant was transferred into a new tube and lyophilised. Metabolites were resuspended in HPLC grade water, diluted as necessary, and injected for analysis. LC-MS/MS was performed as per [49]. Metabolites were loaded onto either a Luna Phenyl-Hexyl column (50 mm × 1 mm × 5 μm particle size) (Phenomenex, Torrance, CA, USA) or a Synergi Polar-RP (reversed phase) 80 Å column (50 mm × 1 mm × 4 μm particle size) (Phenomenex) using a Nexera UHPLC system (Shimadzu, Kyoto, Japan). Sample loading and elution were as described [49]. Metabolites were eluted into a Q-TRAP 5500 mass spectrometer (SCIEX, Framingham, MA, USA) operated in targeted MRM mode. Data files were imported into Skyline (v. 4.1.0.18169), and peak areas manually integrated. Statistical analysis was performed in Metaboanalyst (v.4.0).

### 2.6. Biofilm Assays

Biofilm formation was assessed as per [49]; briefly, overnight cultures were diluted to an OD_600_ of 0.1 in 24-well flat-bottom cell culture plates containing fresh MH, fresh minimal MCLMAN media [50], and both MH and MCLMAN supplemented with 5% (*v*/*v*) CE. Plates were incubated for 48 h under microaerophilic conditions. Non-adherent, non-biofilm cells were removed by washing with 1 mL of sterile PBS. A total of 1.2 mL of fresh medium supplemented with 0.01% (*w*/*v*) 2,3,5-triphenyltetrazolium chloride (TTC) was added to each well and incubated at 37 °C under micraerophilic conditions for 72 h. TTC was removed and the wells were air dried. Bound dye was dissolved using 20% acetone/80% ethanol and absorbance A_590_ was measured. 

### 2.7. Analysis of Lipid A by MALDI-TOF MS

Preparation of lipid A was performed as per [49,51] with minor modifications. Briefly, 10 mg wet-weight cells were solubilized in 4:3 70% isobutyric acid/1 M ammonium hydroxide and boiled at 100 °C for 45 min. Insoluble material was removed by centrifugation for 15 min at 2000× *g*, following which supernatants were diluted 1:1 with ultrapure water and lyophilised. Extracts were washed twice with methanol and reconstituted in 2:1:0.25 chloroform/methanol/water (CMW). Aliquots were spotted onto stainless steel MS targets 1:1 with a norharmane matrix (10 mg/mL [Sigma, St. Louis, MO, USA] in 2:1:0.25 CMW). Matrix-assisted laser desorption ionization time-of-flight (MALDI-TOF) MS mass spectra were acquired on a Bruker UltrafleXtreme (Bruker Daltonics, Billerica, MA, USA) operated in negative ion reflectron mode across a mass range of 1000–2500 *m/z*, and a laser frequency of 2000 Hz using 40% global intensity. For each growth condition, *n* = 5 spectra were analysed using MALDIquant v.1.22.2 [52]. Spectra were square-root transformed, smoothed using the Savitzky–Golay method, and the SNIP algorithm used for baseline correction. Spectra were normalised to total ion count, aligned with a mass tolerance of 0.002, averaged between replicates for each condition with intensity binned within a mass of 0.002, and peaks selected from ions with a signal/noise ratio greater than 15. Statistical significance was calculated using values from replicate spectra, using student’s *t*-test (assuming unpaired samples and 2-tailed variance).

### 2.8. Phenotypic Assays

Polymyxin B sensitivity was measured using Etest^®^ strips (Biomérieux, Marcy-l’Étoile, France). First, 0.2 mL culture grown as above was spread on MH agar with or without 5% (*v*/*v*) CE, and an Etest^®^ strip was added to the centre. Plates were incubated for 48 h at 37 °C under microaerophillic conditions and the concentration at which no growth was observed equalled the minimum inhibitory concentration (MIC). Resistances for ciprofloxacin, ampicillin, and amoxicillin/clavulanic acid were measured using antimicrobial susceptibility discs (Thermo Scientific) on plates as grown above with the zone of inhibition measured in mm. Resistance against oxidative stress was measured by exposure to 5 mM H_2_O_2_ for 30 min with % survival calculated by plate count [49]. Motility was assessed as described [49] using both semi-solid MCLMAN and MH media (with and without 5% CE) in 0.4% agar. Plates were inoculated with 2 μL of an overnight biphasic culture (OD_600_ 0.5) and incubated for 48 h at 37 °C under microaerophilic conditions and motility was measured by diameter of bacterial spread. Western blotting using anti-CadF antiserum was performed as described in [53].

## 3. Results

### 3.1. Proteomics of C. jejuni Response to Growth in 5% and 100% CE

Quantitative proteomics by LC-MS/MS was undertaken using a TMT label-based approach post-trypsin digest to compare relative abundances of proteins from *C. jejuni* NCTC11168 grown in MH medium with those grown in MH supplemented with 5% filter-sterilised CE (5% CE) and in 100% CE alone (100% CE). Data comparing MH controls with 5% CE were acquired from *n* = 4 growth (biological) replicates across two separate LC-MS/MS experiments (two biological replicates per LC-MS/MS experiment). LC-MS/MS identified peptides corresponding to 1386 unique *C. jejuni* proteins, with 58 of these identified by a single peptide alone removed from further analysis. We therefore confidently quantified 1328 proteins (with ≥two peptides per protein from at least one of two LC-MS/MS experiments), representing 81.8% of the *C. jejuni* NCTC11168 translated genome of 1623 genes (Data S1). A similar comparison was performed for additional MH controls and 100% CE acquired from *n* = 3 biological replicates run across separate experiments. Here, LC-MS/MS identified peptides from 1326 proteins, with 22 identified by only a single peptide. Hence, for this analysis, we quantified 1304 proteins, representing 80.3% of the predicted *C. jejuni* proteome (Data S1).

Log_2_ (fold change) and −log10(*p*-value) were arrayed by volcano plots (Figure 1A), which showed the global effects of 5% and 100% CE on the *C. jejuni* proteome. We next converted log_2_ values to *n*-fold change for ease of further analysis. Proteins that were significantly altered in abundance were defined as those with a mean >±1.5-fold change (<0.67 and >1.50-fold reproduced across a minimum of 2/4 or 2/3 biological replicates for each comparison, respectively) and with *p* < 0.05 (Appendix A). Alignment of the data for each individual protein across the two comparisons showed high correlation (*r* = 0.7462; Figure 1B) and thus conservation of similarly altered proteins, with the magnitude of change appearing to be greater following growth in 100% CE compared with growth in 5% CE (Figure 1C). This could also be shown by examining mean fold change for all proteins with positive (>1.0) regulation (mean 1.299-fold in 5% CE, 1.522-fold in 100% CE; *p* < 0.0001) or those deemed altered in abundance (>1.5-fold; 1.870-fold in 5% CE, 2.166-fold in 100% CE; *p* < 0.0001), while for those with negative regulation (<1.0), we saw no difference in the 5% CE comparison (0.79-fold in 5% CE, 0.78-fold in 100% CE; not significant) and a small yet significant difference for those altered in abundance (<0.67-fold; 0.58-fold and 0.53-fold in the 5% and 100% CE comparisons, respectively, *p* < 0.05). In summary, 5% CE supplementation largely altered the same contingent of proteins as growth in 100% CE, with 100% CE leading to larger fold changes (Figure 1C).

We next performed STRINGdb v.12.0 (https://string-db.org) analysis to identify functional relationships between similarly regulated proteins observed in the 5% and 100% CE comparisons with MH controls (Appendix A). For all four analyses (proteins with increased/decreased abundance in 5% and 100% CE comparisons), we observed significant enrichment of functional pathways. For proteins elevated in abundance in 5% CE, we observed five clusters (flagellar motility, iron uptake and antioxidants, antibiotic resistance, nutrient transport and metabolism, and the *pgl* protein *N*-glycosylation system [Appendix A]), all of which were also observed in the 100% CE comparison that contained two additional clusters (peptidoglycan and cell shape, and lipooligosaccharide [Appendix A]). For those proteins decreased in abundance, STRINGdb identified three clusters in the 5% CE comparison (metabolism, virulence and antigenicity, and protein secretion and translation [Appendix A]); we observed only two clusters in the 100% CE comparison (metabolism and protein translation [Appendix A]). Collectively, these results suggest that CE, irrespective of whether it is used as a supplement or directly for growth, induces a signature response in *C. jejuni* and our data provide testable phenotypes for further validation.

### 3.2. CE Increases C. jejuni Flagellar Motility and Induces Biofilm Formation 

We next wished to examine how changes in the proteome attributed to CE exposure influenced *C. jejuni* phenotypes. Given that flagellar motility was a strongly enriched functional cluster positively associated with growth in both 5% and 100% CE, we firstly specifically examined the relative abundances of all known *C. jejuni* proteins involved in flagellar motility and observed many to be elevated in abundance, while only a few were considered reduced (Figure 2A). We next examined how CE influenced this phenotype; we employed 5% CE in two separate growth models, MH medium as conducted for proteome analysis and minimal, defined MCLMAN medium. The addition of 5% CE significantly increased motility as determined by migration through semi-soft agar, irrespective of base medium (Figure 2B). Previous studies have shown that CE induces biofilm [47]. As expected, cells exposed to 5% CE had significantly elevated biofilm-forming capabilities (Figure 2C), again irrespective of medium. For *C. jejuni* grown in MCLMAN, which were almost incapable of biofilm-like growth, the addition of 5% CE increased biofilm formation ~seven-fold, with CE inducing a ~two-fold increase in MH medium. 

### 3.3. Growth in CE Leads to Increased C. jejuni Antibiotic Resistance 

We observed STRINGdb clusters for proteins involved in antibiotic resistance in experiments comparing *C. jejuni* growth in both 5% and 100% CE compared to MH controls. Proteomics analysis for all known proteins involved in antibiotic resistance (Figure 3A) highlighted several proteins that were elevated in abundance including Cj0607–Cj0608 (1.68-/2.35-fold and 1.85-/2.47-fold in 5% and 100% CE comparisons, respectively) and many of the tripartite efflux system proteins (CmeABC/CmeDEF), although in some cases these did not reach the 1.5-fold threshold, particularly in the 5% CE comparison. Despite this, all showed a positive trend when *C. jejuni* was exposed to CE. We next examined *C. jejuni* susceptibility to a range of antibiotics (Figure 3B). Exposure to CE resulted in increased resistance to ciprofloxacin, consistent with changes to Cme efflux proteins. CE also significantly increased ampicillin resistance, which correlated with increased abundance of the Cj0299 (*bla*-OXA-61) beta-lactamase (1.87-/1.44-fold; *p* = 0.006 in 5%, n.s. in 100%, likely due to fewer peptides quantified and no identification in 1/3 experiments for 100% CE). Finally, CE resulted in increased resistance to the beta-lactam/beta-lactamase inhibitor combination, amoxicillin, and clavulanic acid (Figure 3B). 

Resistance to the antimicrobial peptide polymyxin B potentially relies on both efflux and interaction with the phosphate moieties of lipid A, amongst other targets [54]. *C. jejuni* grown with CE demonstrated increased polymyxin B resistance (Figure 4A). We observed increased abundances of LpxH, which catalyses the fourth step in lipid A biosynthesis (producing mono-phosphorylated lipid X; 1.55-/2.31-fold in 5% and 100% CE comparisons, respectively) and the lipid export protein MsbA (2.56-fold in 100% CE) (Figure 4B). All other proteins were largely unaltered. We next employed MALDI-TOF MS to determine whether CE induced structural changes to lipid A (Figure 4C). We observed increased abundances of two lipid A structures—mono-phosphorylated lipid A and phosphoethanolamine (pEtN)-mono-phospho-lipid A (Figure 4D)—with both of these structures showing a greater than two-fold increase in abundance in 5% CE, commensurate with reduced, but not significant, abundances of several di-phosphorylated structures (Figure 4E). We observed no significant change in the lipid A pEtN transferase EptC or the lipid A kinase LpxK. Therefore, from our data, it appears likely that the increased production of lipid X and transport to the periplasm by MsbA accounts for the increased proportion of unmodified and pEtN-modified mono-phosphorylated lipid A.

### 3.4. CE Induces Iron Uptake and Leads to Increased Oxidative Stress Resistance in C. jejuni

The most striking observation in our analysis of 5% and 100% CE was the large cluster of induced proteins involved in iron acquisition (Figure 5A), including the ferrous iron transporter FeoB, enterobactin iron-siderophore uptake proteins CeuD/CeuE, heme uptake transport proteins ChuACDZ, putative ferric iron uptake proteins CfbpA and CfbpC, and iron uptake-associated proteins Cj0177–Cj0178 and Cj1658–Cj1663. Many of these are repressed by the ferric uptake regulator (Fur), which itself was present at reduced abundance (Figure 5A). This suggests CE contains components that sequester iron, for example, the presence of serum transferrin. Reduced iron availability also correlates well with our observations of largely reduced abundances of Fe–S cluster proteins following growth in CE (Appendix A). Since iron is intimately linked to ROS, we next examined oxidative stress and antioxidant proteins (Figure 5A). Here, we observed a significantly increased abundance of catalase KatA during growth in CE (1.54-/2.76-fold in 5% and 100% CE comparisons, respectively). Therefore, we next assayed for *C. jejuni* survival in 5 mM H_2_O_2_ following growth in MH medium with and without 5% CE and found significantly increased resistance to oxidative stress, consistent with elevated KatA (~1.8-fold increase in survival [Figure 5B]). 

### 3.5. CE Results in Changes to C. jejuni Nutrient Transport 

Proteins involved in nutrient transport were strongly enriched in STRINGdb analysis of CE growth. We next aligned the proteomics data for *C. jejuni* grown in 5% and 100% CE compared with MH controls to determine how CE influences nutrient transport at the individual protein level (Figure 6A). We observed elevated CE-associated abundances of the dipeptide transporter CptA (Cj0204; 2.24-/5.62-fold in 5% and 100% CE comparisons, respectively), lactate transporter LctP (1.81-/2.43-fold), serine transporter SdaC (1.51-/1.84-fold), and the C4-DCTs DcuB (2.58-/4.09-fold), and DctA (1.64-/2.64-fold). Concurrently, we observed significantly reduced abundances of the aspartate/glutamate transport proteins Peb1A and Peb1C (0.56-/0.37-fold, 5%/100% CE compared with MH controls), the amino acid binding proteins CjaA (0.43-/0.49-fold) and CjaC (0.48-/0.57-fold), and the α-KG permease KgtP (0.66-/0.67-fold). To determine whether these and other proteomics-based changes resulted in an altered nutrient uptake phenotype, we performed targeted metabolomics by LC-MS/MS on intracellular metabolites from *C. jejuni* grown in MH with and without 5% CE (raw data are found in Data S2) and arrayed the proteomics data for proteins involved in metabolism (Appendix A).

Metabolomics analysis showed *C. jejuni* grown in 5% CE displayed increased intracellular levels of lactate and serine, consistent with the data for LctP and SdaC (Figure 6B). We also observed elevated intracellular levels of proline, which we could not robustly correlate with the abundance of the transporter PutP, as it was only quantified accurately in 2/4 biological replicates in the 5% CE comparison (1.26-fold mean abundance increase in 5% CE), and not at all in 100% CE. Additionally, we observed decreased intracellular aspartate, consistent with data acquired for Peb1A. Metabolomics also showed reduced intracellular abundance of the dipeptide cystine, which was consistent with reduced abundance of TcyP (Cj0025c; 0.40-fold in 5% CE compared with MH control; not statistically quantified in 100% CE). Alignment with proteomics data for proteins involved in the catabolism of these nutrient sources showed that elevated lactate and serine corresponded with significantly reduced abundances of the enzymes LutABC and SdaA (Appendix A), both of which may relate to the lack of iron availability for Fe–S centres (required for catalysis by SdaA and LutA/LutB). We were unable to quantify α-KG; however, we noted significantly reduced abundances of OorABCD consistent with reduced α-KG uptake for use as an electron donor. Proline uptake by PutP is linked to PutA, which was also reduced in abundance in the presence of CE (Appendix A). Accumulation of these carbon sources via elevated abundance of their transporters, yet reduced abundance of their catabolic enzymes, suggests CE contains elevated levels of serine, proline, and lactate, and that *C. jejuni* may be capable of storing these nutrients while depleting other substrates. 

### 3.6. CE Alters C. jejuni Depletion of Specific Substrates from Culture Supernatants

Given the changes to nutrient transporters and intracellular metabolites, we next attempted to determine whether growth in the presence of 5% CE influenced nutrient depletion from MH medium corresponding to our observed changes in intracellular abundances. The addition of CE had no effect on the rate of aspartate or glutamate depletion from culture supernatants (CSN), with both almost completely exhausted from the medium by 24 h growth (Figure 7A). For serine, we observed significantly less depletion from CE-supplemented medium at 4 h compared with MH alone (~62% of control compared with ~49% of control in MH alone), which is likely caused by elevated starting levels of serine in CE compared with MH alone, irrespective of increased SdaC-mediated uptake. By 24 h growth and beyond, serine was similarly nearly completely depleted from both media (Figure 7A). We observed maintenance of CSN proline levels from MH medium with 5% CE compared with MH alone at 24, 48, and 72 h growth; no depletion from the medium was observed at 4 and 24 h, while in MH alone, proline was nearly completely exhausted by 24 h. These data, combined with our observations for intracellular proline abundance, suggest that CE is highly proline-rich and proline may continue to be utilised after other preferred substrates are exhausted. Asparagine was depleted from both media with no significant differences in rate and there was no evidence of glutamine uptake, which is consistent with strain NCTC11168 not encoding a gamma-glutamyltranspeptidase (GGT) and with previous observations [49].

Under specific conditions, succinate can be secreted from *C. jejuni* into the CSN by the C4-DCT DcuB (2.58-/4.09-fold in 5% and 100% CE compared with MH controls) and then re-acquired via DctA (1.64-/2.64-fold) (Figure 6A). Both DCTs (and DcuA, for which we did not obtain reliable quantitative data) are also able to transport fumarate and aspartate into the cell. Intracellular metabolomics confirmed increased levels of fumarate in *C. jejuni* grown in the presence of 5% CE (Figure 7B). Additionally, *C. jejuni* contains two enzyme complexes capable of reducing fumarate to succinate (FrdABC and MfrABE). FrdABC also catalyses the reverse oxidation of succinate to fumarate. Proteomics confirmed that FrdABC is either unaffected or significantly reduced in abundance during growth in CE, while MfrABE is significantly elevated (MfrA, 1.63-/1.95-fold in 5% and 100% CE comparisons, respectively; MfrB, 1.59-/1.62-fold; and MfrE, 1.52-/1.69-fold). Despite this, intracellular succinate levels are not significantly increased in MH with 5% CE (Figure 7C). Analysis of CSN showed that *C. jejuni* grown in both MH with 5% CE and MH alone secreted succinate, with CSN succinate levels peaking at ~500% of uninoculated control in MH and ~300% in MH with 5% CE at 4 h (Figure 7D). These high CSN levels of succinate were sustained for a significantly longer duration when *C. jejuni* was grown with 5% CE; at 24 h, succinate was almost entirely depleted from the CSN of MH cultures but was maintained at ~200% of control in medium with 5% CE (Figure 7D). By 48 h, succinate had been almost entirely depleted. These data collectively suggest that CE is fumarate-rich and limits oxygen availability, resulting in Mfr up-regulation and succinate production, prolonged succinate excretion, and subsequent re-uptake.

*C. jejuni* can utilize a limited range of dipeptides, most likely as nitrogen and sulphur sources [49,55,56]. Other bacteria can also acquire mammalian or ‘meat’-associated peptides, such as carnosine (β-alanine-histidine) and its metabolic product anserine (β-alanine-3-methyl-histidine), as well as carnitine (with derivatives including acetylcarnitine). Intracellular metabolomics of *C. jejuni* identified large increases in carnosine and anserine following growth in MH with 5% CE (Appendix A). The addition of 5% CE to defined minimal MCLMAN medium replicated these data and allowed quantification of carnitine and acetylcarnitine (Appendix A). We next examined CSN to see whether this apparent uptake was associated with commensurate depletion from the growth medium. For carnitine and acetylcarnitine, we observed no evidence of depletion from CE-supplemented medium. We did identify and quantify anserine in MH medium in the presence of 5% CE and observed some depletion relative to control (to ~65% of the uninoculated control at 72 h growth; Appendix A). No anserine could be identified in MH medium without supplementation. Carnosine was identified in CSNs from both MH and MH with 5% CE, and in both cases, we observed depletion (to ~40% of control in MH, and to ~65% in MH with 5% CE).

### 3.7. CE Alters Abundance of C. jejuni Virulence-Associated Proteins

STRINGdb analysis of the *C. jejuni* proteome response to growth in CE showed several clusters associated with virulence traits, including virulence factors, known adhesins, and immunogens. We observed consistently elevated levels of four proteins, the secreted factors, CdtBC and CiaI (Cj1450), and the tyrosine kinase outer membrane protein, Cjtk/Omp50. Conversely, an array of membrane-, virulence-, and immunogen-associated proteins were consistently reduced in abundance during growth in CE. These included Peb1A, Peb1C, Peb2, Peb3, Peb4A, CjaA, CjaC/HisJ, Pal/CjaD, and CadF (Figure 8A). Western blotting using anti-CadF anti-serum confirmed these data in both *C. jejuni* exposed to 5% CE in defined minimal MCLMAN and MH media (Figure 8B). Proteins involved in chemotaxis were also altered by exposure to CE, with several transducer-like proteins (Tlps) present at reduced abundance including Tlp3, Tlp8, Tlp9/CetA, and CetB (Appendix A). Finally, proteins involved in peptidoglycan (PGN) biosynthesis, modification, and assembly, and thus maintenance of helical cell shape (Appendix A), were also altered by growth in CE, with CE inducing increased abundances of the biosynthetic enzymes MurDEF, the lytic transglycosylases MltG, RlpA, and Slt (Cj0843c), the cell shape determinant Csd1 (Cj1087c), and the PatA *O*-acetyltransferase responsible for transporting acetate into the periplasm for PGN modification.

We also saw consistent elevation in CE growth-associated abundance of proteins representing the *pgl N*-linked glycosylation system (Appendix A). The Pgl system can glycosylate proteins and generate a free oligosaccharide that is thought to be crucial in resistance against osmotic stress. We firstly examined whether an increased requirement for *N*-glycosylation might correlate with increased abundance of known glycoproteins; here, we observed significantly higher numbers of glycoproteins present in CE growth at >1.5-fold elevated abundances than present at <0.67-fold abundance (15 and 28 up-regulated proteins for 5% and 100% CE comparisons, respectively, versus 10 and 9 proteins down-regulated; Appendix A). This suggests an increased demand for *N*-glycan to satisfy the increased abundance of glycoprotein targets. Since UDP-*N*-acetylglucosamine (UDP-GlcNAc) is the building block for *N*-glycan biosynthesis, as well as PGN and lipid A, we assayed intracellular levels of UDP-HexNAc, which showed that growth in CE leads to significantly reduced abundance of this metabolite (Appendix A), consistent with an increased requirement for synthesis of these three compounds and aligning with our proteomics data for elements of all three pathways. 

## 4. Discussion

*C. jejuni* typically causes human disease following transmission via the food chain. Survival on food products, most notably poultry for human consumption, is therefore the primary route of infection and a critical junction at which intervention could reduce pathogenesis. Little is known about how *C. jejuni* responds at the molecular level in the different environments that are encountered between poultry and human hosts. One proposal has been to observe the organism in various models of food product growth, including CE [28,47]. While the nutritional composition of CE is not fully defined, studies have shown a matrix enriched in proteins and amino acids, and depleted in carbohydrates, compared with general media [29]. Here, we aimed to understand how the proteome of *C. jejuni* responds during growth in CE. We found that CE elicits widespread changes to the proteome that were generally reflected in corresponding phenotypes, including nutrient acquisition, motility and biofilm, remodelling of lipid A, and resistance against oxidative stress and antibiotics.

Many studies have examined the food microbiology of *C. jejuni* in poultry, often with a focus on antimicrobial resistance gene prevalence and production of virulence factors [6,7,57]. Several interventions are utilised worldwide to limit *C. jejuni* in poultry and supermarket products post-cull, ranging from prophylactic antibiotic treatments to chemical treatment on meat products [58,59,60]. For the CE utilized in this study, we are unable to say whether, and if so which, chemical or antibiotic treatments were applied during production, and hence whether such compounds may be present. However, over several years, we have observed some batch-to-batch variations in CE collected from different sources; these variations were highlighted by, for example, differences in abundance changes associated with antibiotic and foreign compound efflux. For this study, we employed a single large batch of CE from an ‘organic’-labelled butchery; however, the final composition or presence of additives remains uncertain.

CE induced changes to flagellar proteins and increased *C. jejuni* motility and biofilm formation, consistent with previous reports on CE-induced biofilm growth [47]. Motility and biofilm have been linked previously in this organism [61,62]. Our data suggest increased motility is not related to the number of flagellin structural units, as neither FlaA nor FlaB were altered in abundance. Evidence of a relationship between motility and biofilm has been established for flagellar mutations in the *flhA* and *pflA* genes [63], which suggest flagellar-mediated adherence is required for biofilm initiation, with motility increasing the rate at which this can occur. We observed elevated FlhA abundance, as well as similarly altered abundances of the motor stator proteins MotA and MotB that regulate flagella rotational force and are required for chicken colonisation [64]. Further support for our data was observed for FliW, which was the most induced flagellar protein in both 5% and 100% CE comparisons. Deletion of *fliW* results in loss of motility and reduced biofilm mass, as well as decreased ability to colonise chickens [65]. FliW acts as an additional FlaA regulator, largely via interactions with CsrA, which itself is able to repress *flaA* expression [66]. Expression of *C. jejuni csrA* in an *E. coli* Δ*csrA* mutant restores both motility and biofilm formation [67]; here, we observed an elevated abundance of CsrA, consistent with these studies and the CE-associated motility and biofilm phenotypes. Flagellar motility is also at least partially regulated by the c-di-GMP binding regulator CbrR [68]. c-di-GMP is a second messenger associated with biofilm and *cbrR* mutants also display hypermotility [69]. Here, CbrR was reduced in abundance in both 5% and 100% CE but did not quite reach the fold-change cut-off we employed (0.73-/0.71-fold). Several factors influence biofilm formation in *C. jejuni*, including aerobiosis, the presence of extracellular DNA, and the availability of specific substrates [62,63,70,71,72], including fumarate that we observed in increased abundance during growth in CE. Collectively, our data suggest that CE increases motility which confers improved flagellar-mediated adherence for biofilm initiation.

Antimicrobial efflux is considered the major resistance strategy employed by *C. jejuni*, yet the organism maintains only two known tripartite efflux systems; the major CmeABC and minor CmeDEF complexes that are involved in resistance to all classes of antibiotics [73]. We observed elevated abundances (although not always reaching magnitude thresholds, particularly in 5% CE) of all six proteins belonging to these complexes. We also observed CE-induced abundances of the macrolide export proteins Cj0607/Cj0608 (MacB); however, we did not test for erythromycin resistance. Increased ampicillin resistance was observed in *C. jejuni* grown with CE. While efflux is a major means of penicillin resistance, *C. jejuni* also encodes a β-lactamase (Cj0299, *bla*-OXA-61 [74]) that was elevated in abundance in 5% CE (1.87-fold). Aligning with our result, increased expression of *cj0299* is required for ampicillin resistance and this expression is mediated by a promoter-region single-nucleotide mutation [75]. Efflux may play a role in polymyxin B resistance; however, many mechanisms are known, and the specific target of this compound is phosphorylated lipid A, resulting in destabilized lipid A cross-links and thus reduced outer membrane integrity [54]. Analysis of *C. jejuni* lipid A showed significantly increased abundances of mono-phosphorylated and pEtN-mono-phosphorylated lipid A, consistent with increased antimicrobial peptide resistance. These changes appear independent of the pEtN transferase EptC (no change in abundance), suggesting that increased production of mono-phosphorylated lipid A drives these structural changes. Supporting this was the increased abundance of LpxH, which produces mono-phosphorylated lipid X during lipid A biosynthesis, MsbA that exports lipid A across the inner membrane, and smaller but reproducible increases in GnnA/GnnB that form the GlcN3N precursor (2,3-diamino-2,3-dideoxy-D-glucose) disaccharide lipid A backbone [76,77]. Both pEtN and GlcN3N are associated with increased polymyxin B resistance.

CE appeared to induce iron limitation as we observed increases in many proteins related to iron and iron-siderophore acquisition. CE may therefore contain plasma proteins, such as serotransferrin, that effectively sequester free iron. Further support for this was the identification and quantitation of several proteins (e.g., the ChuABCDZ heme acquisition system and CfbpA/CfbpC) that we previously only observed under iron starvation induced by deferoxamine; furthermore, the *C. jejuni* CE proteome aligns closely with the signature observed during growth with this iron chelator (unpublished data). The apparent iron-starved state of growth in CE was reflected by reduced abundances of almost all known *C. jejuni* Fe–S cluster proteins, and lack of iron is known to limit Fe–S cluster protein biosynthesis [78]. The relationship between iron availability and ROS stress is generally well understood in *C. jejuni* and is largely determined by the Fur and PerR regulators, both of which repress iron-regulated genes in the presence of iron [45,79]. Catalase *katA* is iron-repressed via these regulators, such that iron starvation leads to elevated KatA, as observed here. The abundance of Fur was reduced (significantly but at a magnitude just >0.67), which is consistent with our observations. We observed no changes in the PerR-regulated AhpC antioxidant protein, nor changes in PerR itself, suggesting the effect on iron and oxidative stress observed is driven primarily by iron starvation via Fur. Increased KatA abundance has also been shown to subsequently confer increased resistance against oxidative stress [80], as we observed for *C. jejuni* grown in CE. 

As for *C. jejuni* growth in deoxycholate [49], changes in nutrient transport protein abundances largely correlated with the relative intracellular abundances of their corresponding substrates. CE induced large changes in SdaC (serine) and LctP (lactate), suggesting both substrates are enriched in CE. Serine and lactate have been associated with *C. jejuni* infection [34,39], as well as inducing catabolic repression of transporters and enzymes involved in the use of other carbon sources [81]. This correlated with our observations of, for example, reduced abundances of KgtP (σ-KG), Peb1, and Peb3 (aspartate) that negatively regulate catabolite repression [81]. Catabolite repression alone, however, cannot explain the CE-associated proteome signature identified here; unlike in [81], we observed strongly increased abundances of DcuB (succinate export) and CstA (dipeptides), as well as likely increased proline transport via PutP. Furthermore, our data suggest that serine and lactate catabolism may be impeded as the serine dehydratase SdaA and components of the lactate dehydrogenase-like LutABC are Fe–S cluster proteins that are down-regulated due to CE-induced iron limitation. This suggests that the increased uptake of these nutrients may facilitate carbon storage and as discussed, we observed an increased abundance of the ‘carbon storage’ regulator CsrA. Our nutrient uptake data confirmed that *C. jejuni* utilizes a specific order of preference for amino acids [32,33]. *C. jejuni* grown with CE utilized proline at an apparently slower rate than when grown in MH alone, with CSN proline levels maintained at 24 h growth in CE. *C. jejuni* will deplete serine, aspartate, and glutamate prior to proline, and serine represses the use of other substrates [81]. Since we observed elevated intracellular proline and prolonged maintenance of CSN proline levels, we suggest that CE contains high levels of proline. While we performed no absolute quantitation of amino acid abundances, in the absence of *C. jejuni*, peak values of uninoculated media suggested an ~20-fold increase in proline in MH with 5% CE. Secondly, this additional proline (as the least preferred carbon source) is depleted later as other nutrients are exhausted, particularly serine, which then removes the catabolite repression of PutP [81]. Despite this, we were surprised that proline was not completely exhausted by 48–72 h (depletion to 15–25% of uninoculated time-matched controls). This favours a model in which CE provides large amounts of proline and that proline utilisation is regulated to maintain survival following the depletion of other preferred carbon sources in this environment.

Catabolite repression by serine and lactate is linked to intracellular succinate accumulation [81]. We observed no change in intracellular abundance of succinate in *C. jejuni* grown in CE, and both MH-only and 5% CE cultures showed evidence of considerable succinate export. Prolonged maintenance of CSN succinate levels (to 24 h) was seen in 5% CE, consistent with elevated DcuB abundance. We noted consistently increased CE abundances of MfrABE (Cj0437–Cj0439) that reduce fumarate to succinate [42]. This suggests CE is fumarate-rich, consistent with elevated intracellular abundance, with MfrABE also known to confer more efficient growth in the presence of fumarate [42]. Additionally, our MfrABE data suggest that CE results in a more oxygen-limited environment than MH alone, as *mfrA* expression is induced by low oxygen. Our data also correlate with observations of a relationship between MfrABE and oxidative stress, with Δ*mfrA C. jejuni* compromised for *katA* expression, catalase activity, and showing dysregulated iron homeostasis [82]. The DcuB DCT is coupled to fumarate reduction by FrdABC in the cytoplasm, while MfrABE is periplasmic [40,42]. We saw largely reduced (or unaltered) FrdABC abundances in the presence of CE. Therefore, catabolite repression and initial accumulation of succinate likely result in DcuB-mediated export, while increased fumarate from CE drives MfrABE. Periplasmic succinate derived from MfrABE and DcuB is then likely excreted from the cell. Increased intracellular fumarate may be driven by DctA, which is also responsible for the re-uptake of succinate [37], and increased abundance, as well as reduced FrdABC activity in the direction of fumarate reduction, is observed in CE. These data collectively show that CE is a fumarate-rich, oxygen-limited environment that induces the export and subsequent uptake of succinate to sustain the survival of *C. jejuni*. 

Our data align well with a study that examined metabolites in frozen/thawed chicken compared with fresh meat [83], with freeze/thaw resulting in increased surface levels of organic and amino acids, and the purine derivative hypoxanthine, for which we also saw increased abundance in *C. jejuni* grown in CE (Appendix A). We also saw preliminary evidence that *C. jejuni* may be able to import ‘meat’-associated dipeptides. Carnosine and anserine are present at relatively high levels in the skeletal muscle of vertebrates, although anserine varies markedly between animals, with poultry being considered ‘high’, beef intermediate, and pork the lowest [84,85]. We observed high intracellular abundances of both carnosine and anserine, as well as carnitine and acetylcarnitine. Examination of uptake data was, however, inconclusive as to whether this is transport mediated by *C. jejuni*; there was evidence for uptake of carnosine and anserine with some CSN depletion occurring between 4–72 h, but not reaching more than ~50% of uninoculated controls. The rate of depletion was similar for both MH and 5% CE, which does suggest some uptake as uninoculated CE contains ~1300-fold more carnosine and ~1500-fold more anserine than MH, and at no time point in either medium were dipeptides exhausted. We saw no evidence for uptake of carnitine or acetylcarnitine. Conversely, these dipeptides may bind non-specifically to the *C. jejuni* outer membrane, despite several washing steps prior to cell lysis for metabolomics analysis. This could also explain the low-level CSN depletion we observed as bacterial cell numbers grow. Alternatively, the *E. coli* HisJ homolog (Cj0734c CjaC in *C. jejuni*) binds carnosine, albeit weakly, and this may be a mechanism for transport in *C. jejuni* [86]. The uptake of dipeptides in *C. jejuni* is at least partly conferred by the carbon starvation protein homolog CstA [55] and the peptide transporter CptA (Cj0204) [56]. We observed a heavily induced abundance of CptA in CE-exposed *C. jejuni*, while CstA was reduced. Δ*cstA* were compromised for the transport of di- and tripeptides containing alanine, histidine, and glycine [55], while CptA can transport a range of dipeptides, particularly those containing cysteine for sulphur acquisition [56]. An alternative sulphur transporter, the cystine dipeptide DCT-like TcyP/Cj0025c [49], was reduced in abundance during CE growth, suggesting cysteine and cysteine-containing peptides are readily available in CE. Given the concentration of meat dipeptides in poultry, it is plausible that *C. jejuni* uses them as a nitrogen source; however, further work is required to confirm this. 

CE resulted in the reduced abundance of proteins involved in adherence and immune generation. Several of these, including CadF and the Peb antigens, have been positively associated with chicken colonisation [13,87], suggesting any molecular signal detected by *C. jejuni* for their expression is not necessarily found in CE. Alternatively, it is plausible that some in vivo signals may lower the production of immunogens to reduce host immunity; however, any mechanisms behind this remain unknown. Some secreted factors were elevated in abundance, but it remains unclear whether this is due to increased synthesis or reduced secretion. CiaI has been linked with chicken colonisation [88], while elevated abundances of the cytolethal-distending toxin subunits CdtB/CdtC in a food-like environment is potentially of major concern [89]. We observed changes in proteins involved in lipid A, PGN, and *N*-linked glycosylation that suggest CE induced their rate of biosynthesis, and this aligned with a reduction in UDP-HexNAc levels, correlating with its usage (as UDP-GlcNAc) as the building block for these pathways. Increased abundances of Pgl proteins, including a moderate increase in the PglF rate-limiting step [90], correlated with increased abundances of many target glycoproteins, including some with high glycan site occupancy [91], yet no obvious functional relationships beyond those discussed (e.g., CmeABC efflux proteins) could be discerned. Increased *N*-glycan biosynthesis could also be associated with elevated free oligosaccharides and hence, improved defence against osmotic stress [92]. Deletion of *pglB* results in both oxidative and osmotic stress resistance defects [23]. Changes in PGN proteins were largely confined to the Mur biosynthetic proteins; however, we also observed changes to the cell shape determinant Csd1 (Cj1087c [93]), the PatA acetate transporter (associated with PGN *O*-acetylation [94]; however, no changes to acetylase/deacetylase PatB/Ape1 were observed), and lytic transglycosylases, most notably Slt (Cj0843c), which generates anhydrosylated PGN [95]. Increased abundances of lytic transglycosylases likely result in increased anhydrosylated PGN terminal sugars, reduced glycan chain length, and increased recycling of PGN during growth in CE [96,97]. Structural analysis of PGN under different growth conditions, including CE, will ultimately determine how changes to such proteins contribute to the cell shape phenotype.

## 5. Conclusions

*C. jejuni* growth in CE resulted in widespread physiological change. CE induced *C. jejuni* biofilm formation, which appeared to be related to flagellar motility for attachment to CE-associated particles. CE altered antibiotic resistance, lipid A, and nutrient transport. Our data suggest CE generates an oxygen- and iron-limited environment that is rich in organic and amino acids compatible with *C. jejuni* metabolism. CE resulted in elevated intracellular serine and lactate that confer catabolic repression; however, we observed no increase in intracellular succinate. This is likely due to oxygen limitation driving succinate export. *C. jejuni* exposed to CE showed prolonged uptake of proline and exported succinate. Our work shows how *C. jejuni* responds to growth in a food-based model system. We also provide evidence of meat dipeptide (carnosine and anserine) transport, although further work is required. This study demonstrated that *C. jejuni* can adapt to poultry surfaces that are enriched in nutrient sources for which the organism is highly metabolically adapted. Our work will assist in identifying interventional targets to minimise or eradicate *C. jejuni* from the food chain.

## Figures and Tables

**Figure 1 microorganisms-12-00860-f001:**
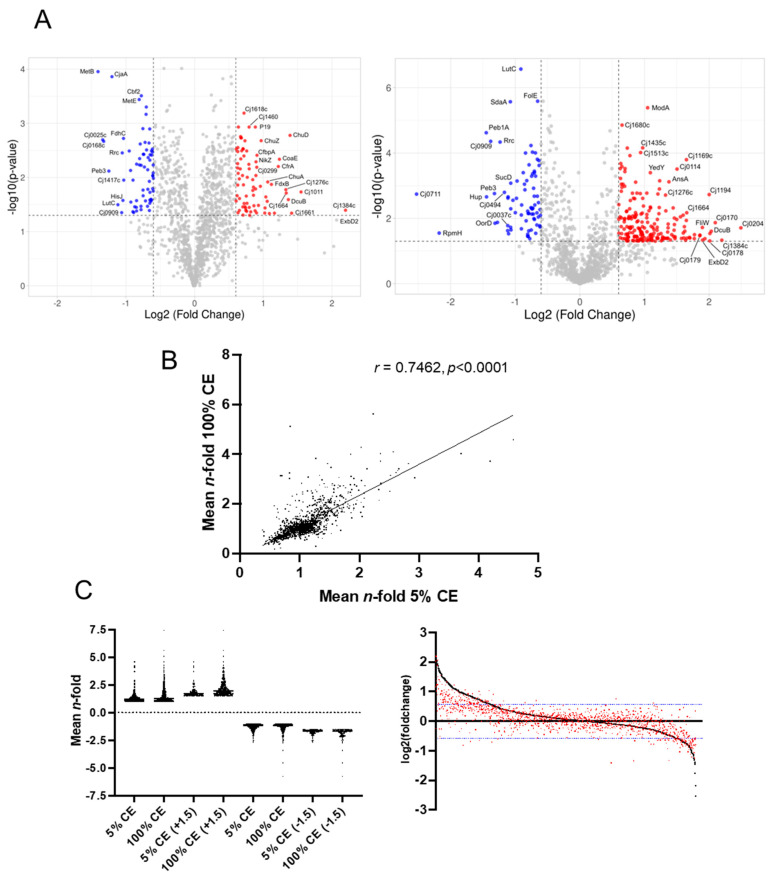
Growth in the presence of CE induces profound changes in the *C. jejuni* proteome. (**A**) Volcano plots of significant protein abundance changes (blue, reduced; and red, increased abundance) for left, growth in MH medium supplemented with 5% CE and, right, growth in 100% CE, compared to MH-only controls. (**B**) Correlation plot shows concordance of proteome abundance changes shared between 5% and 100% CE compared with MH control; (**C**) (**left**) Distribution of abundance changes shows that the magnitude of induced change is larger for shared proteins following growth in 100% CE compared with 5% CE; (**right**) plot shows abundance changes ranked according to observation in 100% CE compared to MH-only control (black) with corresponding change for each protein in 5% CE compared with MH-only control shown in red. Blue lines indicate significant log_2_ fold changes.

**Figure 2 microorganisms-12-00860-f002:**
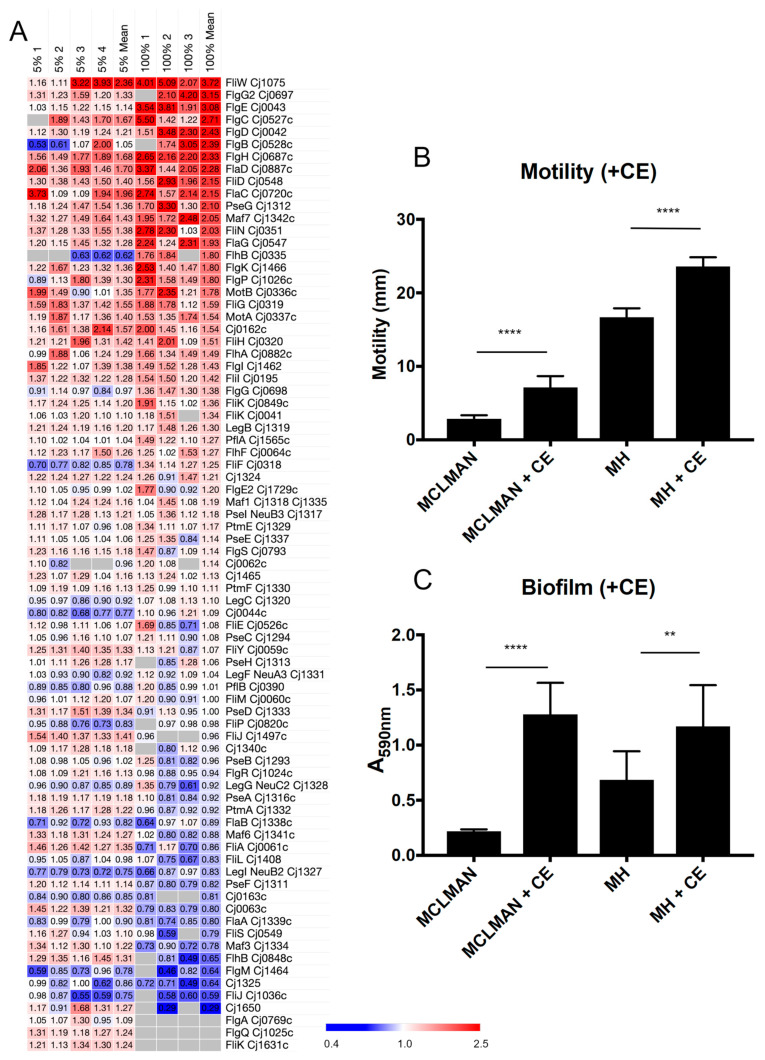
Growth with CE leads to altered *C. jejuni* motility and biofilm formation. (**A**) Heat map showing replicate (MH + 5% CE [left; *n* = 4] and 100% CE [right; *n* = 3] compared with MH-only control) and mean *n*-fold data for proteins involved in flagellar motility (grey indicates not quantified); (**B**) Motility assays as measured by migration (mm) through semi-solid defined MCLMAN and MH agar; (**C**) Biofilm assay as measured by absorbance at 590 nm; **, *p* < 0.01; ****, *p* < 0.0001.

**Figure 3 microorganisms-12-00860-f003:**
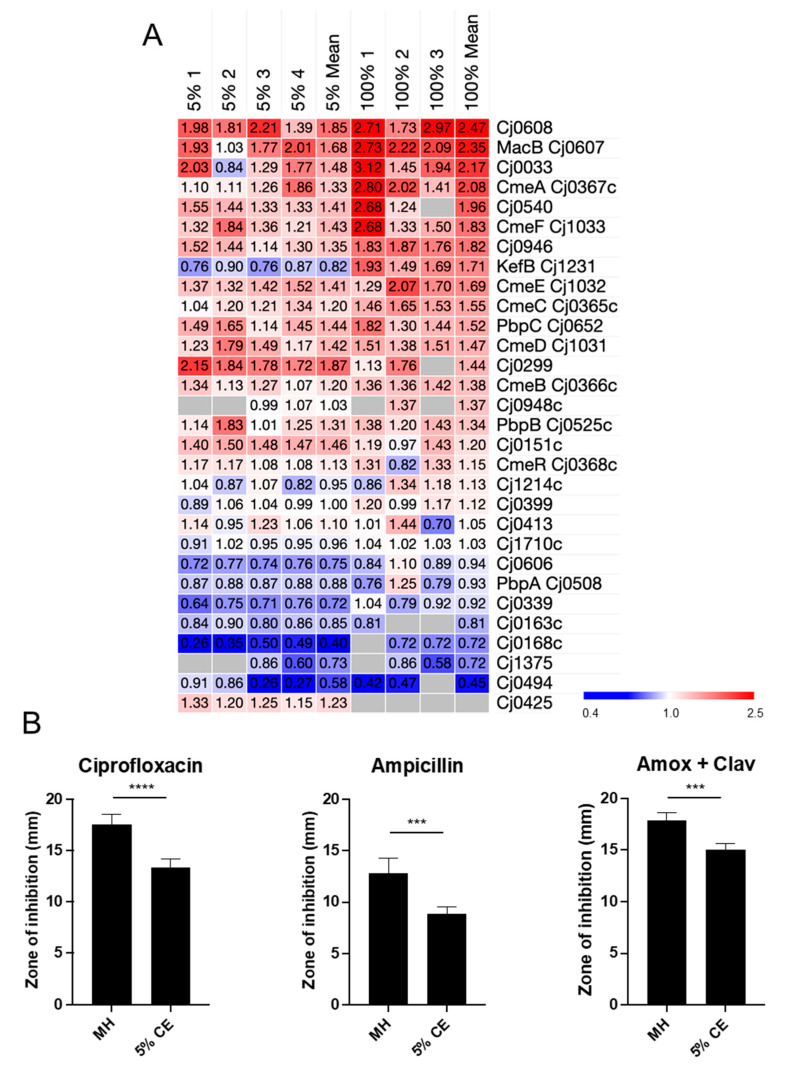
Growth with CE leads to altered *C. jejuni* antibiotic resistance. (**A**) Heat map showing replicate (MH + 5% CE [left; *n* = 4] and 100% CE [right; *n* = 3] compared with MH-only control) and mean *n*-fold data for proteins involved in antibiotic resistance (grey indicates not quantified); (**B**) Antibiotic resistance assays as measured by zone of inhibition (mm) using antibiotic disks; ***, *p* < 0.001; ****, *p* < 0.0001.

**Figure 4 microorganisms-12-00860-f004:**
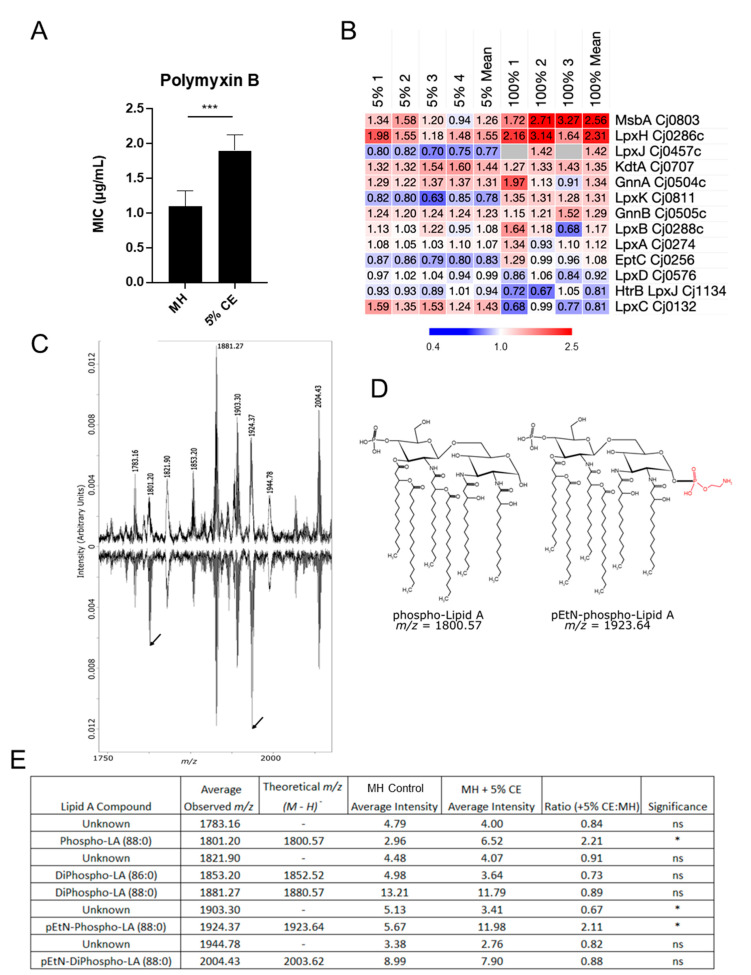
Growth in CE increases abundance of mono-phosphorylated lipid A variants. (**A**) Polymyxin B resistance assay as measured by MIC; (**B**) Heat map showing replicate (MH + 5% CE [left; *n* = 4] and 100% CE [right; *n* = 3] compared with MH-only control) and mean *n*-fold data for proteins involved in biosynthesis and modification of lipid A (grey indicates not quantified); (**C**) MALDI-TOF MS spectra of lipid A from *C. jejuni* grown in (upper) MH only and (lower) MH supplemented with 5% CE (arrowheads indicate peaks at 1801.2 and 1924.37 *m/z*; (**D**) Chemical structures of mono-phosphorylated lipid A (red, position of pEtN); (**E**) Table from *n* = 5 biological replicates of MALDI-TOF MS peak intensities *, *p* < 0.05; ***, *p* < 0.001, ns, not significant.

**Figure 5 microorganisms-12-00860-f005:**
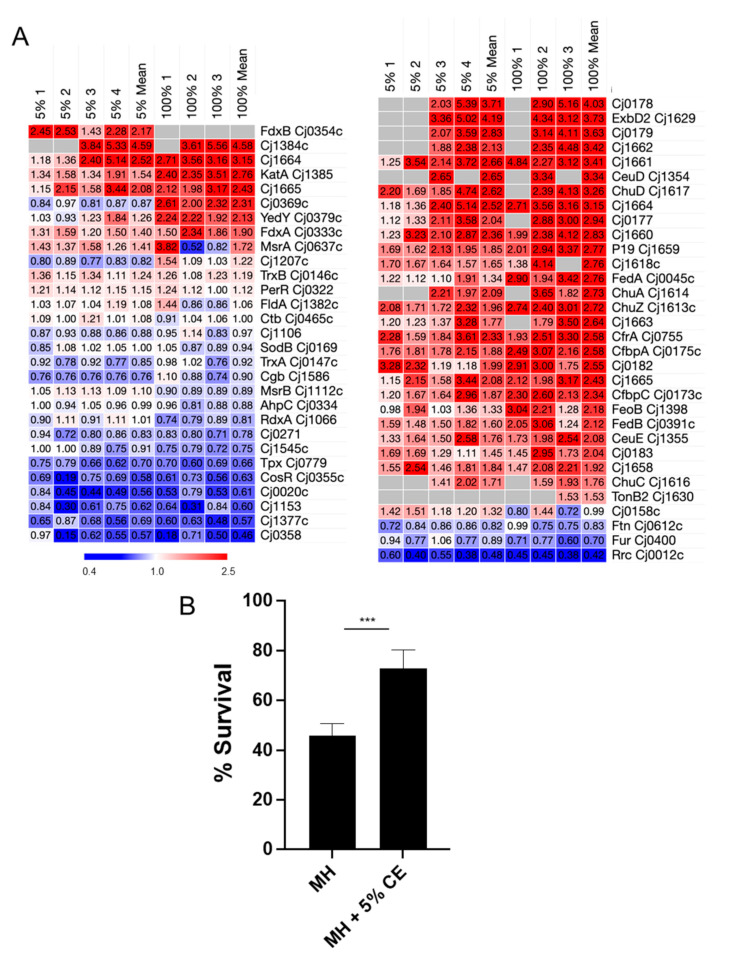
Growth with CE increases *C. jejuni* resistance against oxidative stress. (**A**) Heat maps showing replicate (MH + 5% CE [left; *n* = 4] and 100% CE [right; *n* = 3] compared with MH-only control) and mean *n*-fold data for proteins involved in (**left**) oxidative stress and antioxidant functions, (**right**) iron and heme acquisition; (**B**) Oxidative stress resistance assays measured by % survival of starting cells following 30 min exposure to 5 mM H_2_O_2_; ***, *p* < 0.001. Grey indicates not quantified.

**Figure 6 microorganisms-12-00860-f006:**
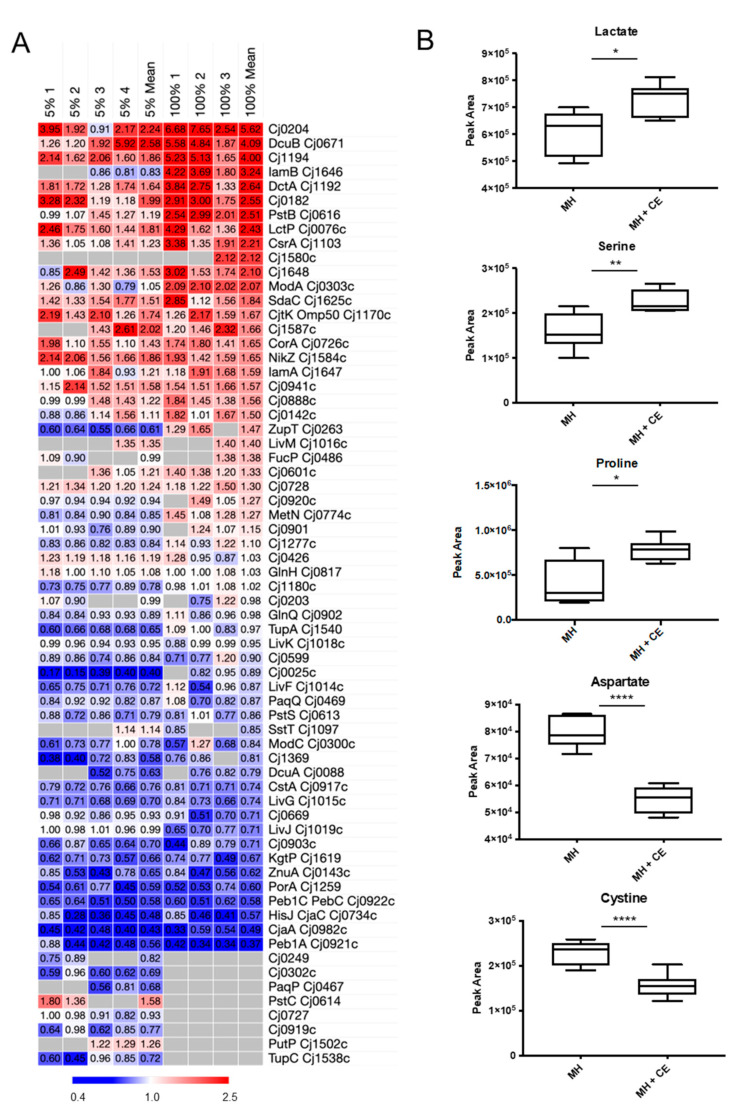
Growth in CE leads to altered *C. jejuni* nutrient transport. (**A**) Heat map showing replicate (MH + 5% CE [left; *n* = 4] and 100% CE [right; *n* = 3] compared with MH-only control) and mean *n*-fold data for proteins associated with nutrient transport (grey indicates not quantified); (**B**) Intracellular metabolite abundances as determined by targeted LC-MS/MS. From the top: lactate (transported by LctP), serine (SdaC), proline (PutP), aspartate (several including Peb1A), and cystine (Cj0025c); *, *p* < 0.05; **, *p* < 0.01; ****, *p* < 0.0001.

**Figure 7 microorganisms-12-00860-f007:**
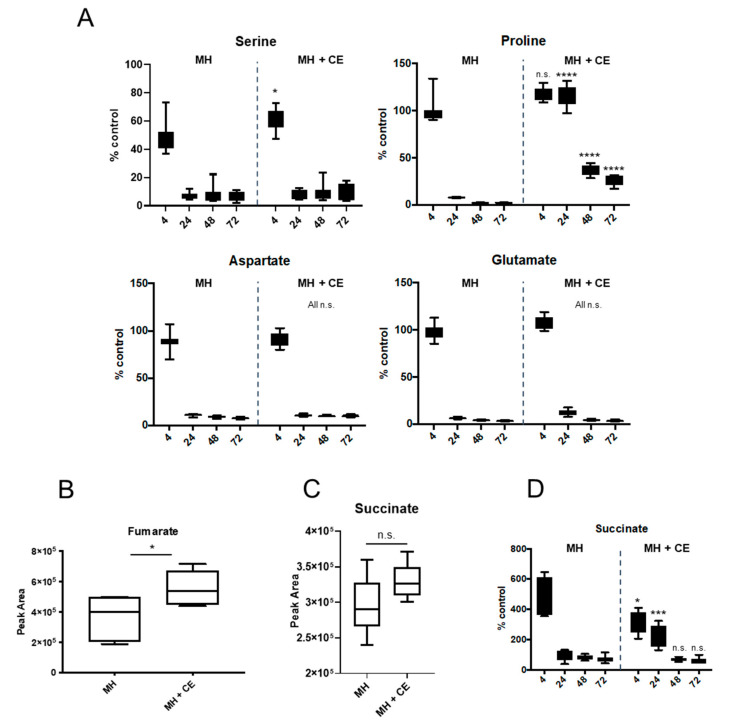
Metabolomics of culture supernatant (CSN) and intracellular levels of *C. jejuni* substrates assayed by targeted LC-MS/MS. (**A**) CSN amino acids at 4–72 h growth; (**B**) Intracellular fumarate; (**C**) Intracellular succinate; (**D**) CSN succinate at 4–72 h growth. Box and whisker plots were based on peak area values determined as a % of the corresponding non-inoculated control (MH and MH + 5% CE) at each time point (Data S3). *, *p* < 0.05; ***, *p* < 0.001; ****, *p* < 0.0001), n.s., not significant; all statistical analyses were performed between replicate % control values across time points, i.e., comparing % control value in MH at each time point with MH + CE at the corresponding time point.

**Figure 8 microorganisms-12-00860-f008:**
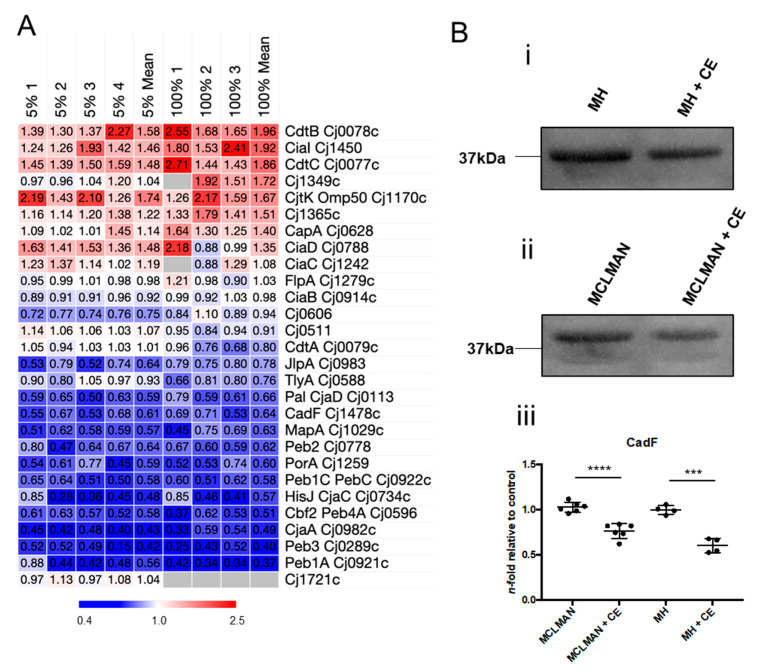
Growth in CE leads to *C. jejuni* proteome changes related to virulence and antigenic proteins. (**A**) Heat map showing replicate (MH + 5% CE [left; *n* = 4] and 100% CE [right; *n* = 3] compared with MH-only control) and mean *n*-fold data for proteins functionally associated with virulence and known antigens (grey indicates not quantified); (**B**) Validation of reduced abundance of CadF in media supplemented with 5% CE from *n* = 5 and *n* = 4. Western blots (MCLMAN and MH media, respectively (**i**,**ii**), supplemented with 5% CE); (**iii**) quantification by densitometry; ***, *p* < 0.001; **** *p* < 0.0001.

## Data Availability

Proteomics data presented in the study have been deposited to the ProteomeXchange Consortium via PRIDE with the dataset identifier PXD051724.

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
