# Peer review of "Multi-Omics of Campylobacter jejuni Growth in Chicken Exudate Reveals Molecular Remodelling Associated with Altered Virulence and Survival Phenotypes"

_microorganisms, 2024, doi:10.3390/microorganisms12050860_

Round 1

Reviewer 1 Report

Comments and Suggestions for Authors

Dear authors

Thanks for your effort for this manuscript “Multi-Omics of Campylobacter jejuni Growth in Chicken Exudate Reveals Molecular Remodelling Associated with Altered Virulence and Survival Phenotypes”.

However, certain comments should be considered during revision;

  1. Line 15, the abbreviation “C. jejuni” should be mentioned after the full name of the bacterium. Line 43, Campylobacter jejuni should be followed by abbreviation (C. jejuni).
  2. Line 43, a recent reference should be added. The same for lines 47-49.  Lines 44-47, too long statement!
  3.  The names of genes should be written in a correct way.
  4.  Line 98, the abbreviation “FADH” should be defined.
  5.  Line 124, the aim of the study containing references [48]!
  6. The source of the used strain should be mentioned.
  7. Lines 121-131, the aim of the study should not include any expected results.
  8. Line 139, the locality (a local commercial poultry butcher?) of the used bacterial strains should be provided. Butchers could be replaced by slaughter houses.
  9. Line 253, why the sensitivity to polymyxin B?!
  10. Lines 608-611, references should be provided.
  11. Line 614, CE, once the abbreviation has been mentioned for the first time, it should be mentioned all over the manuscript.
  12. Some recommendations could be added to the conclusion section. The conclusion could be more concise.

Best wishes

Author Response

Please see the attachment for full response to reviewers.

Reviewer 2 Report

Comments and Suggestions for Authors

Overall, the article provides a comprehensive analysis of C. jejuni response to chicken exudate using a multi-omics approach, contributing valuable insights to the field of food microbiology and food safety. I do have only some suggestions to make it even more comprehensible in terms of expression

  1.  
    1.  

    2.  

    3.  
Comments on the Quality of English Language
    1.  
    2. Line 28 "CE also led ... - This statement Is not very clear to the readers that are not familiar. It should specify that the increased intracellular abundances of serine, proline, and lactate were correlated with the increased abundances of their respective transporters.

    3. Line 29 "Analysis of ...." - This sentence again is not very simple to follow. It should specify whether the prolonged retention of proline and succinate refers to their increased levels in the culture supernatants or their reduced uptake by the bacteria.

    4. Line 36"CE is both oxygen ..." - The subject-verb correlation is incorrect. It should be "CE is both oxygen and iron limiting and provides evidence of factors required for phenotypic remodeling to enable C. jejuni survival on poultry products."

    5. Line 64 "Despite a limited genome C......" - This sentence is overly long and complex. It should be broken into smaller sentences so that one that tries to follow could have a better understanding.

      1. Line 565 "the tyrosine kinase outer membrane protein ...." - Consider adding a comma after protein.

      2. Line 616 "Here, we aimed ..." - Consider rephrasing for clarity: "Our aim was to understand how the proteome of C. jejuni responds during growth in CE".

      3.  

      4.  

Author Response

(The authors gave the same response as above.)
